# Urinary Biomarkers in Monitoring the Progression and Treatment of Autosomal Dominant Polycystic Kidney Disease—The Promised Land?

**DOI:** 10.3390/medicina59050915

**Published:** 2023-05-10

**Authors:** Camelia Pana, Alina Mihaela Stanigut, Bogdan Cimpineanu, Andreea Alexandru, Camer Salim, Alina Doina Nicoara, Periha Resit, Liliana Ana Tuta

**Affiliations:** 1Nephrology Department, Faculty of Medicine, “Ovidius” University of Constanta, 900470 Constanta, Romania; 2Medical Semiology Department, Faculty of Medicine, “Ovidius” University of Constanta, 900470 Constanta, Romania; bogdan.cimpineanu@365.univ-ovidius.ro (B.C.); alina.nicoara@365.univ-ovidius.ro (A.D.N.); 3Nephrology Department, Constanta County Emergency Hospital, 900601 Constanta, Romania; 4Emergency Department, Constanta County Emergency Hospital, 900601 Constanta, Romania; salimcamer@yahoo.com; 5Faculty of Medicine, “Ovidius” University of Constanta, 900601 Constanta, Romania; cupi.periha@yahoo.com

**Keywords:** ADPKD combined with one of the following: urinary biomarkers, diagnosis, risk stratification, progression, treatment, urinary exosome, non-coding RNAs, proteomics, TKV, specific biomarkers, disease severity

## Abstract

Autosomal dominant polycystic kidney disease (ADPKD) is the most common genetic kidney disease, and it leads to end-stage renal disease (ESRD). The clinical manifestations of ADPKD are variable, with extreme differences observable in its progression, even among members of the same family with the same genetic mutation. In an age of new therapeutic options, it is important to identify patients with rapidly progressive evolution and the risk factors involved in the disease’s poor prognosis. As the pathophysiological mechanisms of the formation and growth of renal cysts have been clarified, new treatment options have been proposed to slow the progression to end-stage renal disease. Furthermore, in addition to the conventional factors (PKD1 mutation, hypertension, proteinuria, total kidney volume), increasing numbers of studies have recently identified new serum and urinary biomarkers of the disease’s progression, which are cheaper and more easily to dosing from the early stages of the disease. The present review discusses the utility of new biomarkers in the monitoring of the progress of ADPKD and their roles in new therapeutic approaches.

## 1. Introduction

Autosomal dominant polycystic kidney disease (ADPKD) is a genetic disorder characterized by the uncontrolled formation and growth of renal cysts due to a mutation occurring in one of the two polycystin genes, PKD1 and PKD2 [1,2] and, much more rarely, by two other recently identified genes, GANAB [3] and PMM2 [4]. The disease affects between 4 and 7 million patients worldwide, and it is found in 7–15% of patients on chronic hemodialysis. The estimation of the prevalence of ADPKD in the general population is difficult, with highly variable reports, ranging from 1.44:10,000 [5] to 25:10,000 [2]. This is mainly due to the significant proportion of asymptomatic patients, for whom there is no simple, fast, and inexpensive screening test. The prevalence of the disease in Europe is 2.7:10,000, according to a recently published meta-analysis, affecting both sexes equally [6].

Two proteins encoded by PKD1 and PKD2, polycystin-1 and polycystin-2, are involved in cyst formation. These proteins are located in the primary cilia of the distal tubular epithelium, acting as a receptor/channel complex [5] and playing an important role in maintaining the intercellular junction, the stabilization of the extracellular matrix, and the transmembrane transport of electrolytes (Ca^2+^) as a partial response to flow-dependent mechanosensory stimuli [7,8]. Deficiencies inf the proteins polycystin-1 and polycystin-2 cause cyst formation, but only in 1–5% of the tubes. However, the disease frequently progresses to ESRD, mainly through the compression/obstruction mechanism of the adjacent tubular structures and of the interstitial tissue, which causes ischemia, inflammation, and progressive fibrosis [9].

The continuous growth of cystic bodies occurs due to uncontrolled cell proliferation and fluid secretion into the cyst lumen [10], secondary to the somatic second-hit mutation [11,12].

Both abnormal cell proliferation and cyst-filling-fluid secretion are cyclic AMP-mediated processes initiated by stimulating the Ras/mitogen-activated protein kinase (MAPK) pathway and by the activation of the cystic fibrosis transmembrane conductance regulator (CFTR) chloride channel [13] and TMEM16A (anoctamin 1), which leads to transepithelial chloride secretion [14] (Figure 1).

Cystic growth is stimulated by major alterations in intracellular metabolism, some of which involve mitochondria, a key storage organ for Ca^2+^, with a major role in cellular Ca^2+^ signaling. Polycystin proteins can directly and indirectly regulate mitochondrial function by regulating calcium signaling, reducing cAMP levels, inhibiting miR-17, maintaining the mitochondrial DNA (mtDNA) copy number, and modulating mitochondrial morphology [16,17]. The disruption of polycystin’s functions induces morphological changes in the mitochondria, which contribute to disruptions in mitochondrial superoxide production, alterations in ERK/MAPK signaling in cyst epithelial cells, decreases in intracellular Ca^2+^ concentrations, reductions in peroxisome proliferator-activated receptor γ coactivator 1α (PGC-1α) expression via calcineurin, Ca^2+^-related molecules p38 MAPK (p38 mitogen-activated protein kinase), and low NOS (nitric-oxide synthase) activity [18]. Reductions in PGC-1α increase the oxidative stress observed in ADPKD and can cause oxidative DNA mutation, which might also explain the high risk of the development of a variety of cancers, especially in the kidneys, liver, skin, and colon, in patients with ADPKD [19].

Angiogenesis may be an important factor in the pathogenesis of ADPKD, especially in cyst growth. The process of angiogenesis begins in the early stages of the disease and is activated by the hypoxia resulting from the growing cysts’ compression by the adjacent renal vessels. The process is mediated by the hypoxia-inducible factor (HIF-1), which binds an angiogenic growth factor, such as VEGF, and promotes endothelial cell proliferation, mitosis, increases vascular permeability and plasma-protein leakiness [20,21]. In addition to hypoxia, the renal expression of Ang-1 and Ang-2 is also known to be stimulated by angiotensin II. Thus, the renin–angiotensin–aldosterone system (RAAS) activation occurs early in ADPKD, which may increase angiopoietin production, affecting renal vascularity and cystic growth [22].

## 2. Materials and Methods

This review summarizes the relevant research investigating urinary biomarkers used in diagnosis, monitoring of progression, and treatment options in ADPKD. A comprehensive systematic search of the online and written research was undertaken using several criteria for inclusion (Table 1). All authors participated in the research process, summing 791 articles from the Medline, PubMed, and Google Scholar databases.

## 3. New Biomarkers in ADPKD

Because ADPKD mainly affects tubular structures with secondary perilesional inflammation, the dosing of inflammatory and tubular lesion markers is of interest to researchers, especially since these markers are relatively inexpensive and easy to dose. Several studies have identified a clear association between these markers and the severity of ADPKD, through assessments using GFR and total kidney volume (TKV) [21,22].

Moreover, the pathophysiological mechanisms responsible for cystogenesis, resulting from the suppression of the activity of polycystin 1 and polycystin 2 (Figure 1), have allowed researchers to identify numerous serum and urinary markers, which are useful for the monitoring of the progression of the disease, including the following: epidermal growth factor (EGF); the inflammatory-molecule-transforming growth factor beta 1 (TGFB1); tumor necrosis factor (TNF); angiotensinogen (AGT), as a part of the renin–angiotensin–aldosterone signaling (RAAS) cascade; vascular endothelial growth factor alpha (VEGFA) and apelin (APLN), as markers for angiogenesis; and the ligand of a G protein-coupled receptor, or vimentin (VIM), which is involved in cytoskeletal organization [8].

The progression of ADPKD was assessed by measuring numerous urinary markers: albumin, as a general kidney-damage marker; IgG, as a glomerular damage marker; N-acetyl-β-d-glucosaminidase (NAG), as a proximal tubular damage marker; heart-type fatty-acid-binding protein (H-FABP), as a distal tubular damage marker; liver-type fatty-acid-binding protein (L-FABP), as a marker of interstitial inflammation and fibrosis, macrophage migration inhibitory factor (MIF), neutrophil gelatinase-associated lipocalin (NGAL), and interleukin-18, as inflammation markers [23].

### 3.1. Inflammation Biomarkers

Although ADPKD cannot be defined as an inflammatory disorder, in the kidneys of adults with ADPKD, the intense infiltration of macrophages, under the influence of MCP-1 secretion by tubular epithelial cells, linked to PKD1 mutations, has been identified in rodent models. In patients with PKD1, but not in those with PKD2 or GANAB mutation, high levels of urinary MCP-1 were observed before the occurrence of significant kidney destruction and fibrosis, which could be considered an early disease-severity marker [24]. Because uMCP-1 secretion is an early event in ADPKD, it should also be considered a target for treatment.

Furthermore, pro-inflammatory molecules, such as TNF-α, osteopontin, and IL-1β were found in the urine and cyst fluid of ADPKD patients, together with the accumulation of infiltrating inflammatory cells, such as macrophages and T cells, in the renal interstitium and urine samples [22].

The macrophage migration inhibitory factor (MIF) is upregulated in cyst-lying cells, and it is mediated by hypoxia-inducible transcription factor (HIF) 1α and cAMP [24,25]. It is responsible for macrophage recruitment, resulting in reduced MCP-1-dependent macrophage accumulation in the cystic kidneys and the subsequent delay of cyst growth in several PKD mouse models [26,27].

In addition, urinary MIF is increased in ADPKD patients and demonstrated a positive correlation with either total kidney volume (TKV) or changes in TKV [28,29]. The MIF inhibitor ISO-1 attenuates HIF-1α- and cAMP-dependent cyst enlargement and can also be used as a therapeutic target to delay cyst growth in ADPKD [27] (Table 2).

The phosphate levels in urine are associated with high levels of fibroblast growth factor 23 (FGF23) and are substantially higher in patients with ADPKD than in other CKD patients. Increases in FGF23 production are mediated by parathyroid hormone (PTH), calcium, and erythropoietin secretion induced by the regional hypoxia-induced upregulation of hypoxia-inducible factor 2α [30].

### 3.2. Tubular Damage Biomarkers

In 2019, Messchendorp et al. [23] identified a urinary-biomarker score by summing the ranking of the tertiles of β2 microglobulin (β2M) and monocyte chemotactic protein-1 (MCP-1) excretion, with a high predictive value for rapidly progressive disease; this value was even higher than that of the Mayo htTKV classification or PKD mutation. Several studies show that β2M reflects proximal tubular damage and that MCP-1 reflects inflammation [31], suggesting that both mechanisms are involved in the progression of ADPKD. Messchendorp et al. showed in a previous study that urinary β2M and MCP-1 excretion were both strongly associated with annual GFR decline in ADPKD after statistical adjustment for conventional risk markers [32]. Less strong, but also significant after adjustment for conventional risk markers, was the association with another proximal tubular damage marker, the hallmark of virtually all proteinuric, toxic, and ischemic kidney diseases: kidney injury molecule 1 (KIM-1) [33]. These markers therefore have the potential to be used as predictive tools in clinical practice and to identify patients who would benefit from long-term therapy with disease-modifying agents.

The AGT/EGF axis, with the dysregulation of EGF receptors (EGFR), seems to play an important role in the molecular mechanisms of ADPKD, and decreased urine EGF (uEGF) indicates tubular atrophy and interstitial fibrosis. Signaling through EGF receptors is essential for the growth, migration, differentiation, and proliferation of cells. In ADPKD, the concentration of EGF in the cystic fluid is significantly decreased, and EGF plasma concentrations and urinary excretion are lower in patients with ADPKD than in controls [34,35]. In addition, urinary EGF levels were significantly correlated with eGFR levels in ADPKD patients, showing that tubular dysfunction may also be associated with renal failure [34].

In patients with ADPKD, the biomarkers related to tubular injuries (KIM-1, β2-microglobulin, NGAL, and L-FABP) had higher urine levels than controls, but with low specificity, and they showed a weak-to-moderate correlation with the current standard of htTKV in ADPKD progression [36].

Ismail et al. [37] showed that serum and urinary levels of NGAL are correlated well with total kidney volume (htKTV), with significantly higher levels in patients with a total kidney volume >1500 cm^3^, suggesting the involvement of this protein in the process of cyst formation and development. Meijer et al. [23] found also a correlation between urinary NGAL levels and tKV, which may indicate that the urinary levels of NGAL may increase only in advanced disease of PKD.

No correlation was found between the distal tubular damage marker H-FABP and ADPKD progression [31], but a recent animal-model study identified a strong correlation between urinary levels of liver-type fatty-acid-binding protein (L-FABP) and the progression of tubulo-interstitial damage in polycystic kidney disease [36]. Therefore, it may be a useful marker for monitoring the progression of PKD.

The levels of osteopontin (OPN), which is expressed by renal epithelial tubular cells and excreted into urine, were reduced in biallelic ADPKD gene (PKD2^−/−^) porcine knockout cells and urinary OPN-excretion levels were lower in patients with rapidly progressive disease than in slow progressors [38] (Table 2).

### 3.3. Angiogenic Biomarkers

Urinary angiotensinogen (uAGT) is a novel biomarker of the activation of the renin–angiotensin system (RAAS) in patients with ADPKD, and it is associated with hypertension, progressive kidney damage, proteinuria, and cardiovascular morbidity and mortality. The RAAS system may also contribute to ADPKD progression by stimulating signaling pathways in renal-cyst cells to promote the growth and malfunction of epithelial transport [8,39].

An immunohistochemical study showed the strong expression of AGT in cyst-lining epithelial cells, as well as nearby compressed tubular epithelial cells, as a result of intrarenal ischemic injury and intrarenal RAAS activation. The plasma and urinary AGT levels were significantly elevated in ADPKD and/or CKD patients when compared with samples from a healthy control group, and they can be used as markers of CKD progression in ADPKD [40]. Urinary AGT/Cr and plasma-renin-activity levels were significantly elevated in hypertensive ADPKD patients. Recent studies showed that urinary AGT/Cr was strongly associated with advanced CKD stages, htTKV, and hypertension [41].

The molecular mechanism of neovascularization involves the secretion of VEGF, which might be triggered by hypoxia in the tubule cells and in cysts during their expansion, restricting the process to areas of cyst growth. In addition, tubular hypoxia inducible factor 1α (HIF-1α) was described as having a strong cyst-growth-promoting effect in ADPKD mice [42]. Urinary levels of the biomarkers associated with renal ischemia, such as HIF-1α, pro-angiogenic gene VEGF, and chemotactic factor for monocyte MCP-1, were elevated in patients with ADPKD; they are correlated with htTKV and may, therefore, be used as biomarkers of kidney injury in ADPKD patients with a mild reduction in GFR [35].

### 3.4. Abnormal-Cell-Metabolism Biomarkers

Along with tubulo-interstitial damage biomarkers, the biomarkers of dysregulated cellular metabolism, which may contribute to cyst formation and expansion, were also studied, some in large clinical trials, as markers of response to treatment. In ADPKD kidney tissues, increased aerobic glycolysis (the Warburg effect), impaired fatty-acid oxidation, and reduced AMP-activated protein kinase (AMPK) activity were detected [43,44]. A large clinical trial, TAME-PKD [45], tested the safety and efficacity of metformin, a regulator of cell metabolism in ADPKD patients, by promoting cellular AMPK activation. The urinary levels of both glycolytic- and oxidative-pathway enzymes and metabolites were detected and measured, in an attempt to establish a correlation with htTKV or eGFR, in order to identify patients with severe disease. Among the glycolytic enzymes, pyruvate kinase M2 (PKM2) and lactate dehydrogenase A (LDHA) excretion were significantly correlated with htTKV, whereas PKM2 excretion was negatively correlated with eGFR, suggesting that PKM2 is an important biomarker of kidney volume in patients with ADPKD before renal-function decline. Furthermore, the urinary levels of succinate—a marker of oxidative metabolism—was positively correlated with eGFR in young male patients, suggesting that a reduction in urinary excretion in males might be associated with more rapidly progressive disease [46,47].

The ability of small-molecule intermediates of cellular metabolism (proteomics and metabolomics) to detect biological effects (i.e., toxicity, or disease) is very useful in the identification of ADPKD in its early stages. Elevated urinary protein levels are observed in ADPKD patients as a result of defects in endocytosis, which disturb the reabsorption of low-molecular-weight proteins by proximal tubular cells.

Recent studies focused on urinary exosome proteomics, seeking to identify a quick and inexpensive method to stratify patients at risk of rapid progression and severe disease, as well as for the evaluation of therapeutic efficacy [47]. The analysis of different proteomic markers can distinguish the degree of severity and rate of progression in ADPKD. For rapidly progressive ADPKD, a broad spectrum of urinary proteins was identified, including the maximal upregulation of the Notch pathway, MAST-4 proteins, and EGFR-kinase, as part of accelerated cell proliferation and the sorting of the nexin vesicular proteins, plakoglobin, desmoplakin, fibronectin, vitronectin, radixin, ezrin, tenascin, and uromodulin (reflecting cyst expansion, cell–cell-adhesion disruption, and cytoskeletal abnormalities). The most highly upregulated proteins in the rapidly progressing group were microtubule-associated serine/threonine kinase (MAST)-4, cytokinesis-associated kinesin-like (KIF), and dynein heavy chain [47,48,49].

Many studies showed that a urinary metabolic profile offers a prognostic performance that is at least equivalent to those of risk-stratification-based imaging methods.

Urinary metabolomics analyses, using nuclear magnetic resonance (NMR) spectroscopy, has opened new perspectives in the assessment of ADPKD’s progression and the detection of the disease in its early stages.

In the DIPAK study, the urinary myoinositol/citrate ratio was an important marker of ADPKD progression in patients with the PKD-1 mutation, similar to the currently used htTKV [50]. Urinary citrate is inversely correlated with advanced stages of ADPKD, and its role in preventing tubular crystal formation by binding calcium may influence cyst formation [51]. Hypocitraturia and acidosis are commonly found in patients with CKD and ADPKD. Myoinositol is a renal osmolyte with a role in protecting renal cells from hyperosmolar stress. The reabsorption of myoinositol in damaged renal cells is decreased, resulting in increased urinary excretion, which explains the prognostic value of urinary myoinositol in the prediction of rapid CKD progression [49].

Podrini et al. [42] performed a comprehensive metabolomics characterization of cells and renal tissues from kidney-specific Pkd1-depleted mice, demonstrating broad metabolic reprogramming in ADPKD, including enhanced glycolysis, reduced Krebs cycle, fatty-acid oxidation, and enhanced fatty-acid synthesis. Significant concentrations of alanine, taurine, 3-hydroxyisovalerate, α- ketoglutarate, acetate, succinate, and other metabolites were found in the urine of mice with early-stage ADPKD. Dekker et al. [52] identified a board spectrum of urinary metabolic markers using quantitative NMR profiling, which were correlated with eGFR and with the future rate of decline in eGFR, with a clear connection between the urinary alanine/citrate ratio and the annual change in eGFR in patients with ADPKD.

In recent years, novel alterations were identified, including the augmentation of kynurenines, polyamines, and indoles, which suggested increased inflammation and microbial dysbiosis, providing insights into PKD pathophysiology that may prove helpful in the diagnosis, monitoring, and treatment of ADPKD [53,54].

Recent research indicates that polycystin-mediated mitochondrial dysfunction and metabolic re-programming contribute to the progression of ADPKD [19,55]. Several studies provide evidence that mitochondrial dysfunction plays a functional role in cystogenesis [56,57]. Kidney cysts in mouse and human ADPKD have increased renal expressions of miR-17, which represses oxidative phosphorylation and FAO by inhibiting peroxisome proliferator-activated receptor alpha (PPARα), a regulator of lipid metabolism [58]. Mitochondrial fatty-acid β-oxidation (FAO) is reduced in cell lines derived from ADPKD mouse models and leads to increase fatty-acid biosynthesis [59,60].

Urinary mitochondrial DNA (mtDNA) copy numbers are correlated with the degree of renal dysfunction and histological damage and may serve as non-invasive biomarkers of renal mitochondrial dysfunction [61]. Moreover, a higher serum mtDNA copy number is associated with less severe disease in ADPKD [62] (Table 2).
medicina-59-00915-t002_Table 2Table 2Urinary biomarkers in ADPKD.TypeBiomarkerFirst Mention in ADPKDRoleInflammation biomarkersMonocyte chemoattractant protein-1 (MCP-1)(Cowley, B.D., Jr., 2001) (Danxia Zheng, 2003) [63,64]A chemotactic factor for circulating monocytes and a proinflammatory activator of macrophages. In ADPKD, MCP-1 promotes cyst growth. Early disease-severity marker in ADPKD 1.Tumor necrosis factor-alpha (TNF-alpha)(Xiaogang Li, 2008) [65]Enhances cyst development in ADPKD.Osteopontin(Cowley, B.D., Jr., 2001) [63]An acidic glycoprotein that is involved in bone remodeling, cell survival, inflammation, and kidney damage. The OPN is decreased in the urine of ADPKD patients.Interleukin 1-beta (IL-1β)(Gardner., K.D., Jr., 1991) [66]Activates two major inflammatory pathways in renal epithelial cells: NF-κB and JAK-STAT. As a result, pro-inflammatory molecules are produced and released, attracting and activating even more infiltrating cells, which aggravate the local injury and ultimately contribute to cyst progression.Macrophage migration inhibitory factor (MIF)(Chen L, 2015) [28]Promotes renal-cyst-cell proliferation in a macrophage-independent manner.Fetuin A(Piazon N,2015) [67]Is a negative acute-phase protein expressed in the brain, liver, bones, kidneys, and respiratory and cardiovascular systems. Regulates insulin signaling, bone resorption, and the precipitation of calciprotein particles. After passing from the plasma through glomerular filtration, it is reabsorbed in proximal tubule cells in the tubule lumen.Tubular damage biomarkersβ2 microglobulin (β2MG)(Dimitrakov D, 1987) [68]The β_2_M rises in glomerular process, leading to proteinuria and competition in the filtered protein load in the reabsorbtion process in the proximal tubule.Kidney injury molecule−1 (KIM-1)(V.Bonventre, 2002) [69]The KIM-1 is a transmembrane protein upregulated in renal tubular cells after ischemic injury, and it is expressed in a variety of kidney diseases, especially in the apical membranes of proximal tub cells. In ADPKD, cyst growth and compression by the neighbouring renal tubules upregulate KIM-1 expression.Neutrophil-gelatinase-associated lipocalin (NGAL)(Davide Bolignano, 2007) [70]High dosages of circulating NGAL protein can attenuate PKD progression, and the process was associated with a protective mechanism involving increased apoptosis, decreased proliferation, and decreased fibrosis and cyst growth.Liver-type fatty-acid-binding protein (L-FABP)(Tsukasa Nakamura MD, 2005) [71]Expressed in renal proximal tubules in humans and involved in free-fatty-acid metabolism; urinary L-FABP levels in ADPKD patients with renal failure are significantly higher than those in ADPKD patients with normal renal function.Angiogenic biomarkersUrinary angiotensinogen (AGT)(Mahmoud Loghman-Adham, 2004) [72]Urinary angiotensinogen and renin excretion increase in ADPKD because of damage to the glomerular filtration barrier, reduced proximal tubular reabsorption, enhanced tubular secretion by intact nephrons, differences in degradation, or ectopic production by cyst-lining epithelial cells.Vascular endothelial growth factor alpha (VEGF-A)(ElsaBello-Reuss, 2001) [73]Increased VEGF-A expression causes glomerular hypertrophy and proliferation of podocytes.Apelin (APLN)(Antonio Lacquaniti, 2013) [74]Aggravates albuminuria by increasing the permeability of podocytes and glomerular endothelial cells, and podocyte injuries are mediated by apelin-triggered ER stress. In ADPKD patients, apelin level is lower than in healthy subjects. Low apelin level is associated with faster kidneyy-function decline and fibrosis.Hypoxia inductible factor (HIF-1)(Wanja Michael Bernhardt, 2007) [75] The HIF-1 promotes cyst growth, primarily due to an increase in chloride-dependent fluid secretion into the cyst lumenAbnormal cell metabolism biomarkersPyruvate kinase M2 (PKM2)(Li Chen, 2015) [28]The PKM-2 is a marker of excessive aerobic glycolysis; previously shown to be elevated in ADPKD preclinical models and in human-ADPKD cystic kidney tissue. Succinate-α marker(Kocyigit I, 2019) [76]Increased succinate might be related to HIF-1α and IL-1β activation.MicroRNA 17 (MiR-17)(Vishal Patel, 2013) [77]Represses oxidative phosphorylation and FAO by inhibiting peroxisome-proliferator-activated receptor alpha (PPARα).Mitochondrial DNA (mtDNA)(Yu Ishimoto, 2017) [19]The mtDNA copy number is correlated with the degree of renal dysfunction and histological damage.

### 3.5. Non-Coding RNAs

In ADPKD urine specimens, miRNAs can be determined with NMR spectroscopy, as well with RT-PCR and hybridization-based profiling methods. The most comprehensively studied miRNA in ADPKD is the miR-17 family, which promotes PKD progression by rewriting cyst metabolism. The overexpression of the miR-17–92 cluster promotes tubular and glomerular cyst formation by directly targeting both PKD1 and PKD2 via miR-17-5p, as well as indirectly, through the inhibition of the PKD2 and PKDH1 genes via the miR-92a-3p’s targeting of the transcription of hepatocyte nuclear factor-1β (HNF-1β), which is necessary in the initiation of nephrogenesis and nephron segmentation in embryonic kidneys.

The upregulation of miR-501-5p in ADPKD cells and tissues activates the mTOR/MDM2 pathway by repressing the phosphatase tensin homolog (PTEN) and the TSC1 gene, resulting in cell proliferation [78]. Increased miR-132-3p in ADPKD was shown to increase mitochondrial superoxide levels by directly repressing the target gene, Foxo3a, indicating that oxidative stress might be one of the key factors in the pathogenesis of ADPKD.

Previously implicated as kidney tumor suppressors, miR-1 and miR-133, as well as miR-223 and miR-199, which were implicated in inflammatory- and fibroblast-cell origin, are dysregulated compared to those in other CKD patients. The miR-182-5p was positively correlated with fibrosis and with the elevated expression of collagen I, collagen IV, and fibronectin in cystic epithelial cells. Furthermore, higher levels of the lymphocyte/monocyte-associated miR-223 and of the fibroblast-enriched miR-199a and miR-199b were observed [79].

Another subset of urinary exosomal miRNAs that could serve as a novel biomarker of disease progression was described recently, in the study by Magayr et al. [17]. They confirmed that miR-192-5p, miR-194-5p, miR-30a-5p, miR-30d-5p, and miR-30e-5p were significantly downregulated in the urine exosomes in both murine and human PKD1 cystic kidneys, and all were significantly correlated with baseline eGFR and ultrasound-determined mean kidney length.

Furthermore, miR-21, miR-193, and miR-214 are known to regulate cyst growth by modulating cyst epithelial apoptosis, proliferation, and interstitial inflammation [80] (Figure 2).

These findings indicate that the dysfunctional miRNA expression in the cystic epithelial cells is an indispensable factor in the development of PKD due to its multiple roles in disease pathogenesis: oxidative stress in mitochondrial metabolism, dysregulated cell proliferation, autophagy and apoptosis, inflammation, fibrosis, cell-to-cell contact via extracellular vesicles, and exosomes [59,80].

The aberrant expression of non-coding RNA in both PKD1 and PKD2 patients provides new insight into the pathogenesis of ADPKD and represents promising biomarkers that can improve diagnostic performance and the evaluation of disease progression, as well as acting as a potential therapeutic target.

Preclinical results showed that the anti-miR-17-5p oligonucleotide RGLS4326 attenuated cyst growth in multiple PKD models [81]. The finding that miR-21 inhibits the apoptosis of cyst epithelial cells, probably through the direct repression of its target gene, PDCD4, makes the manipulation of miR-21 expression by miR-21 inhibitors, such as RG-012, another possible therapeutic approach for polycystic kidney diseases [82].

## 4. Discussion

Autosomal dominant polycystic kidney disease is a disease with complex pathophysiological pathways, in which there is a highly variable rate of progression to end-stage renal disease (ESRD). Large cohort studies also showed an increased incidence of kidney cancer in patients with ADPKD, so early diagnosis is a real challenge, especially in patients with ESRD [83]. The lack of prognostic indicators with which to predict disease progression remains a major clinical problem and a source of concern for both physicians and patients. In recent decades, major progress has been made in the more accurate description of the pathogenic pathways of ADPKD’s development and progression, which has allowed the identification of promising targeted therapies, as well as new biomarkers with potential roles as prognostic indicators. Furthermore, the staging of the disease and the classification of the risk of its rapid progression could be very useful in the more careful monitoring of patients, who could benefit from the optimization of their treatment, if necessary.

The identification of urinary metabolites represented the next step in building a biomarker profile, which is necessary for the assessment of ADPKD. Although the Mayo classification for predicting disease progression in ADPKD is widely used [84], new urinary biomarkers have been shown to be easy to use, inexpensive, and at least as accurate in assessing the disease’s progression, its severity, and responses to treatment [60,67]. Urinary biomarkers may be useful for predicting the variable evolution of ADPKD from cyst growth to ESRD. Some of these markers have been shown to distinguish ADPKD patients from controls with a high degree of accuracy [40,85,86].

Furthermore, some urinary biomarkers have been shown to be useful in distinguishing between gene polymorphisms and pathogenic mutations in the absence of family history [38].

Moreover, studies of molecules targeting specific pathophysiological pathways, starting with drugs that act on the calcium/cyclic adenosine monophosphate (cAMP) pathway and continuing with those that act on the epidermal growth factor (EGF) receptor, AMP-activated protein kinase (AMPK), KEAP1-Nrf2, sphingolipids, and microRNAs, have used urinary markers to monitor treatment response, with specificity and sensitivity comparable to the total renal volume or eGFR [8,16,33,87,88,89,90].

Studies of urinary biomarkers have shown that patients with ADPKD have a unique profile of protein in their urine, which can differentiate them from healthy controls and groups with other kidney diseases.

Abnormal cell metabolism is a considerable source of ADPKD-specific urinary biomarkers. Kistler et al. [91] identified a highly specific urinary proteomic profile for the diagnosis and risk stratification of ADPKD when tested on a cohort consisting of 481 patients with a variety of renal and extrarenal diseases, including AKI. A large number of urinary collagen fragments are downregulated in ADPKD and negatively correlated with htTKV. The downregulation of c-terminal fragments of uromodulin and osteopontin is associated with ADPKD severity.

A proteomic analysis of urinary extracellular vesicles (uEVs) identified more than 30 proteins overexpressed in ADPKD patients’ samples compared to healthy subjects and other non-ADPKD kidney diseases. Among these, periplakin, envoplakin, villin-1 were highly correlated with total kidney volume [92].

Several other potential urinary and plasma biomarkers of ADPKD have recently been reported, including NGAL [70], MCP-1 [64], KIM-1 [69], but these markers proved to be non-specific for ADPKD and are also found in healthy subjects.

## 5. Conclusions

In the wide field of biomarkers, their respective utilities in diagnosis, prognosis, prediction, monitoring, and treatment response in PKD should be distinguished, and further validation of urinary biomarkers is necessary.

In the future, targeted therapies could change the natural history of ADPKD, and urinary biomarkers will be needed for rapid and specific assessment of treatment response, as well as for the selection of eligible patients.

## Figures and Tables

**Figure 1 medicina-59-00915-f001:**
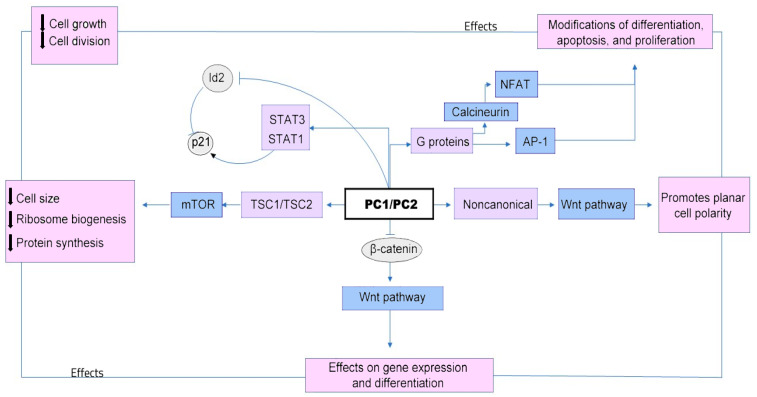
Downregulation of the PC1 and/or PC2 proteins leads to cyst formation and growth through multiple signaling pathways, including mammalian target of rapamycin (mTOR) and members of the pro-inflammatory gene complex, including Janus kinase (JAK)-signal transducers and activators of transcription (STAT), as well as planar cell polarity (PCP), Wnt signaling to some target genes (which promotes epithelial-cell proliferation and cytogenesis), cyclic adenosine monophosphate (cAMP), G-protein coupled receptor (GPCR), cystic fibrosis transmembrane conductance regulator (CFTR), epidermal growth factor receptor (EGFR), mitogen-activated protein kinase (MAPK), cellular Ca^2+^, and the cell cycle [15].

**Figure 2 medicina-59-00915-f002:**
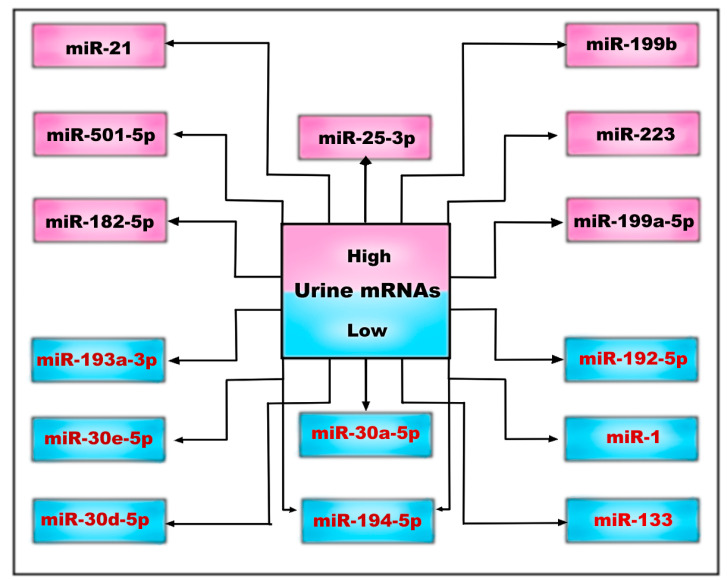
Urinary microRNAs may be low or high, depending on whether they are down- or upregulated in the complex pathophysiological processes of ADPKD.

**Table 1 medicina-59-00915-t001:** Criteria for literature review.

Inclusion Criteria	Exclusion Criteria
Keywords: ADPKD combined with one of the following: urinary biomarkers, diagnosis, risk stratification, progression, treatment, urinary exosome, non-coding RNAs, proteomics, TKV, specific biomarkers, severity	Non-English-language articles
Full texts or review abstracts published between the years 1991 and 2023	Review, case reports, letters
	Clinical trials, preclinical studies, observational studies

## Data Availability

No new data were created.

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
