# Peer review of "Urinary Biomarkers in Monitoring the Progression and Treatment of Autosomal Dominant Polycystic Kidney Disease—The Promised Land?"

_medicina, 2023, doi:10.3390/medicina59050915_

Round 1
Reviewer 1 Report
Overall, this review covers a number of novel biomarkers in monitoring the progress of ADPKD and their therapeutic potential. The manuscript is very well written and informative. Therefore, I only have a minor suggestion to improve the manuscript:
As many biomarkers are shared between CKD and ADPKD. In the manuscript, I would suggest to highlight the specific biomarkers that have been shown to be more specific to ADPKD and can aid in diagnosis and treatment.
Author Response
Thank you for this very good suggestion and i consider that a clarification of this aspect was necessary. That's why a added considerations regarding the specificity for ADPKD of some of urinary biomarkers in discussions section.
Reviewer 2 Report
This study is a systematic review that considers different urinary biomarkers in monitoring the progression and treatment of autosomal dominant polycystic kidney disease (ADPKD).
It is a very good paper based on up to date references (96 of them), involving two very useful figures and a table summarizing the various urinary markers in ADPKD.
The manuscript consists of an introduction (related to the incidence and main pathophysiological mechanisms of ADPKD), a comprehensive overview of new biomarkers in ADPKD, a brief discussion and conclusion.
I will just suggest the authors to add a Material and Methods section with the database they used, key words and criteria for inclusion of the articles in the study.
Author Response
Thank you for this very useful suggestion. I will include a Material and Method section in the article.